# Effect of Ce Addition on Modifying the Microstructure and Achieving a High Elongation with a Relatively High Strength of As-Extruded AZ80 Magnesium Alloy

**DOI:** 10.3390/ma12010076

**Published:** 2018-12-26

**Authors:** Zheng Wang, Jin-Guo Wang, Ze-Yu Chen, Min Zha, Cheng Wang, Shi Liu, Rui-Fang Yan

**Affiliations:** Key Laboratory of Automobile Materials of Ministry of Education & School of Materials Science and Engineering, Nanling Campus, Jilin University, No. 5988 Renmin Street, Changchun 130025, China; wangzheng-jlu@foxmail.com (Z.W.); chenzy1616@gmail.com (Z.-Y.C.); minzha@jlu.edu.cn (M.Z.); chengwang@jlu.edu.cn (C.W.); liushi-jlu@foxmail.com (S.L.)

**Keywords:** AZ80 magnesium alloy, Ce modification, extrusion, microstructure, elongation, strength

## Abstract

Forming magnesium alloys with rare earth elements (La, Gd, Nd, Y, Ce) is a routine method for modifying their microstructure and properties. In the present work, the effect of Ce addition on the microstructure evolution and the mechanical properties of as-extruded Mg-8Al-0.5Zn (AZ80) alloy was investigated. All of the extruded AZ80-*x*Ce (*x* = 0, 0.2, 0.8 and 1.4 wt %) alloys exhibited equiaxed grains formed by fully dynamic recrystallization, and the grain size of the extruded AZ80 alloy was remarkably reduced by ~56.7% with the addition of 1.4 wt % Ce. Furthermore, the bulk-shaped Al_4_Ce phase formed when Ce was first added, with the Ce content rising to 0.8 wt % or higher, and Al_4_Ce particles in both the nano- and micron sizees were well distributed in the primary *α*-Mg matrix. The area fraction of the Al_4_Ce particles expanded with increasing Ce content, providing more nuclei for dynamic recrystallization, which could contribute to the grain refinement. The results of the tensile tests in this study showed that Ce addition effectively improved the room temperature formability of the as-extruded AZ80 alloy, without sacrificing strength. The significantly improved mechanical properties were ascribed to excellent grain refinement, weakened texture strength, an increased Schmid factor, and a reduced area fraction of low-angle grain boundaries, all resulting from Ce addition to the as-extruded AZ80 alloy. The contribution of the nano-Al_4_Ce precipitates on improving the mechanical properties was also discussed in this paper.

## 1. Introduction

Reducing vehicle weight is an important strategy for boosting fuel economy and depressing carbon dioxide emissions. Magnesium and its alloys, as the lightest metallic structural materials, have been of great interest for automotive industries [1]. However, the widespread adoption of Mg alloys has been conspicuously limited by its unsatisfied room temperature formability. This is due to a lack of active slip systems, resulting from the hexagonal close-packed (HCP) crystal structure of Mg alloys [2]. Furthermore, most of the commercial products fabricated from magnesium alloys are nowadays produced with die casting, whereas the use of wrought magnesium alloys is very uncommon, because of its limited formability, despite wrought Mg alloys performing better in terms of mechanical properties than the casting Mg alloys [3]. Therefore, in order to expand the industrial applications, it is essential to enhance the strength as well as the formability and the plastic workability of magnesium alloys, especially the wrought ones [4].

Commercial magnesium alloys can be classified into four categories: Mg-Al, Mg-Mn, Mg-Zn, and Mg-RE (rare earth)-based alloys [5]. AZ series Mg alloys, which are based on the Mg-Al system, have been widely used as commercial wrought Mg alloys, such as the AZ31, AZ61 and AZ80 alloys [6]. Among these wrought Mg alloys, the AZ80 alloy has attracted much attention, owing to the relatively lower cost and more reasonable combination of mechanical properties and corrosion resistance [7,8]. The elements Al and Zn contained in the AZ80 alloy are reported to improve its mechanical properties by solid solution strengthening and precipitation strengthening in the form of a Mg_17_Al_12_ intermetallic compound (*β*-phase) [9]. In the automotive field, AZ80 alloys has been widely used in the manufacturing of automobile wheels and dashboards, and it is essential to find methods to improve the comprehensive mechanical properties, in order to further expand its application.

The addition of alloy elements has long been known as one of the most efficient methods for achieving better mechanical properties of Mg alloys, especially for rare earth (RE) elements, such as La, Gd, Nd, Y, and Ce [10,11,12,13,14,15].

Recently, there have been some investigations on refining the microstructures and improving the mechanical properties of the AZ80 alloy with the addition of RE. For instance, Bu et al. [10] found that the as-extruded AZ80 exhibited a much higher strength and finer grains, due to the mutual effect of the light REs (1.0 wt % La) and heavy REs (0.5 wt % Gd), yet the ductility decreased. Wang et al. [11] investigated the effect of Nd on the as-cast AZ80 alloy, with the addition of 1.0 wt % Nd, and the grain size became much smaller. After T6 heat treatment, the yield strength and tensile strength both remarkably increased. Ren et al. [12] have studied the aging hardening behavior and precipitation evolution of extruded AZ80 with Y addition, ranging from 0.2 wt % to 0.8 wt %. It was reported that the short-term aging softening took place at the beginning of T5 treatment, which resulted in a lower Vickers hardness, and the addition of Y was responsible for this. Moreover, discontinuous precipitation was gradually suppressed with an increasing Y content. 

The effects of adding element Ce to magnesium alloys containing Al and Zn on the microstructure and mechanical performance were also studied. Lü et al. [13] discovered that Ce could achieve a fine morphology, modify the fluidity of the melt, and improve the high temperature tensile properties of the as-cast AZ series Mg alloys. The influence of Ce addition on the microstructure evolution and mechanical properties of extruded Mg-7Sn-1Al-1Zn (TAZ711) alloy have been researched by Kim et al. [14]. The results demonstrated that the grain structure of the as-extruded TAZ711 alloy was composed of fine dynamically recrystallized (DRXed) grains, and coarse unDRXed grains. With increasing Ce content, the particle-stimulated nucleation (PSN) included by the relatively large-sized Ce_3_Sn_5_ phase brought about the increase of the area fraction of the DRXed grains. Meanwhile, the sizes of the DRXed grains became larger, because the fine Mg_2_Sn particles reduced, due to the consumption of Sn by the formation of the Ce_3_Sn_5_ phase. All of these contributed to a decrease in the tensile strength, and an increase in the elongation. Fan et al. [15] verified that with the addition of 2 wt % Ce to the as-cast AZ91D alloy, the finer *β*-Mg_17_Al_12_ and Al_4_Ce strengthening phase appeared. Therefore, the tensile strength and elongation increased by 19.3% and 104.0%, respectively, while the corrosion rate decreased by about 21.8%. However, little attention is paid on the effect of Ce addition on the wrought AZ80 alloy, in spite of the fact that Ce is extensively utilized to modify the microstructural characteristics and mechanical properties of Mg alloys containing Al and Zn elements.

Therefore, in this work, 0.2–1.4 wt % Ce was added into Mg-8Al-0.5Zn (AZ80) alloy to investigate the effect of Ce addition on the microstructural evolution and tensile properties of the as-extruded AZ80-*x*Ce (*x* = 0, 0.2, 0.8 and 1.4 wt %) alloys.

## 2. Materials and Experimental Procedures

Mg-8Al-0.5Zn (AZ80)-based alloys with different Ce contents were fabricated by melting with the mixture of commercial pure magnesium (99.85%), pure aluminum (99.90%), pure zinc (99.90%), and Mg-28 wt % Ce master alloys. The melting procedure was carried out in an electric resistance furnace under the protection of an inert atmosphere with a mixture of CO_2_ and SF_6_ at around 700 °C, and then the melts were poured into a steel die preheated to 200 °C. The dimensions of each cylindrical billet were 94 mm in diameter and 200 mm in length. The chemical compositions of each billet were measured by an optical spectrum analyzer (ARL 4460, Ecublens, Switzerland); each condition has been tested over three times, and for each time, the date result was obtained by measuring the values of 3–4 points on the tested specimens and calculating their average number. The results are presented in Table 1, which shows that the measured compositions were very close to the nominal ones. The cast billets were then machined into cylinders of 90 mm in diameter and 100 mm in length. In order to improve the alloys’ hot deformation performance, a homogenized treatment is necessary to make the non-equilibrium structure of the cast billets uniform in composition by atom diffusion at a high temperature. According to previous experience in our laboratory [16], the billets were homogenized at 415 °C for 20 h, followed by water-quenching. Before extrusion, the billets and the extrusion container were preheated to 430 °C, and then the extrusion was performed at a speed of 0.4–0.6 m/min, and an extrusion ratio of about 32:1. Figure 1 illustrates the schematic diagram of the extrusion process in this work.

Phase analyses were conducted by an X-ray diffractometer (XRD; 18KW/D/Max 2500PC, Rigaku, Tokyo, Japan) with Cu Kα radiation at a voltage of 40 kV at a scanning rate of 4° min^−1^, with the sample tilt angle ranging from 20° to 80°. An optical microscope (OM; Carl ZEISS Axio-Imager-A2m, Göttingen, Germany), a scanning electron microscope (SEM; Carl ZEISS EVO-18, Göttingen, Germany) equipped with an energy dispersive spectrometer (EDS; Oxford INCA-X-Max, England, UK), and a field-emission transmission electron microscope (TEM; JEOL 200KV/JEM-2100F, Akishima, Japan) were used to characterize the microstructures of the extruded samples. In addition, at least eight OM or TEM micrographs for each condition were collected, to obtain the average grain size of the extruded samples, as well as the average size of the second phases by using the Nano Measurer System 1.2 software. The grain structures and microtextures were analyzed by using a SEM (VEGA3 XMU, TESCAN, Brno, Czech) equipped with an electron backscattered diffraction (EBSD) detector (Oxford Aztec-Nordlys-Nano, England, UK). The textures were measured with a step size determined by the grain size, and the Aztec and Channel 5.0 software (Aztec Software Private Limited, Bengaluru, Karnataka) were utilized to manipulate and to collect the date. The metallographic samples for the OM and SEM observations were firstly ground with sandpapers, and mechanical polished with 0.5 μm diamond pastes, and then chemically etched in acetic picric solution for about 30 s. The picric solution was composed of 5 g picric acid, 5 mL acetic acid, 10 mL distilled water, and 80 mL ethanol, of which the picric acid was manufactured by China Taishan Yueqiao Reagent Plastics Company Limited, and the acetic acid and ethanol were manufactured by Beijing Chemical Works. Samples for TEM observations were firstly ground to about 70 μm in thickness, and this was followed by ion thinning for about 2 h via ion beam milling (Leica EM- RES101, Solms, Germany). Specimens for EBSD analyses were electro-polished with a specific commercial AC2 electrolyte solution at 20 V for 25–30 s after mechanical polishing.

Tensile tests were carried out via a tensile testing machine (INSTRON 5869, Norwood, MA, USA) at room temperature under a strain rate of 1.0 × 10^−3^ s^−1^. Tensile specimens with a gauge size of 10 × 3.5 × 1.8 mm were machined, with the load axis parallel to the extrusion direction (ED). Moreover, the values that appeared in this work were the average of at least 3–5 sets of measured dates, and the stress-strain curves presented were good representatives.

## 3. Results and Discussion

Figure 2a–d shows the OM images of the extruded alloys. These micrographs reveal that the grains of the extruded alloys are all equiaxed grains formed by fully dynamic recrystallization (DRX). Further, the grains of Ce-containing alloys are much finer than those of the alloy with no Ce addition, When the Ce content is gradually added from 0 to 1.4 wt %, the grain size of the as-extruded AZ80 alloy decreases obviously from ~34.2 μm to ~14.8 μm.

By analyzing the XRD results of the as-extruded AZ80-*x*Ce (*x* = 0, 0.2, 0.8, and 1.4 wt %) alloys (Figure 3), it can be indicated that there were two phases in the as-extruded AZ80 alloy: primary *α*-Mg and *β*-Mg_17_Al_12_; a new phase, Al_4_Ce, emerged with the addition of Ce. Due to the low content of Zn in these alloys, there were no Zn-containing phases detected. Probably, most of the Zn dissolved into the *α*-Mg matrix uniformly. It is reported that in the AZ series alloys, if the Zn/Al ratio exceeds 1/3, Mg-Al-Zn ternary phases may appear [17]. In this work, Zn/Al ratio was ~1/16, far less than 1/3, and hence showing no Mg-Al-Zn phase. The results also showed that with the increase of Ce addition, the intensity of diffraction peaks corresponding to the Al_4_Ce phase rose, while the intensities of the diffraction peaks of *β*-Mg_17_Al_12_ phase declined. This was due to the consumption of Al during the formation of the Al_4_Ce phase, and thus causing the reduction of the *β*-Mg_17_Al_12_ phase. From the diffraction peaks at the angle of ~33 degrees, ~58 degrees, and ~71 degrees, it can be seen that the intensity of the diffraction peaks corresponding to the *α*-Mg matrix declined, while the full width at half maxima (FWHM) became broader with the increase of Ce content. This was mainly because the grains of the *α*-Mg matrix were greatly refined with the addition of Ce (Figure 2), and a smaller grain size often results in more serious X-ray dispersion and increases the FWHM. The decline of the intensity of the *α*-Mg matrix’s diffraction peaks can also indicate that the texture of *α*-Mg matrix was weakened. Furthermore, it is worth mentioning that the phase peaks of the XRD spectrum shifted, which might be attributed to the changes of the crystal lattice constant caused by the extrusion processes.

Figure 4a–d shows the SEM photographs of the extruded alloys, and Figure 4e–h represents the corresponding magnified SEM photographs of the selected zones A, B, C, and D in Figure 4a–d, respectively. Figure 4i–l are corresponding EDS mapping chemical analyses of Figure 4e–h. The EDS chemical composition tests of the as-extruded specimens show that Al and Ce elements were abundant in the area of some second phase particles. Combined with the results of XRD shown in Figure 3b–d, it can be deduced that these second phase particles may be Al_4_Ce. From SEM micrographs, it can be clearly seen that Mg_17_Al_12_ particles often segregated near the serrated grain boundaries, while they rarely appeared inside the grains, as well as being adjacent to the flat grain boundaries. The reason can be that the serrated GBs will provide more nucleation for the Mg_17_Al_12_ particles [12]. The corresponding backscattered electron (BSE) images of Figure 4a–d are shown in Figure 4m–p, respectively. The Al_4_Ce particles in these images are revealed as bright white, and their area fractions are estimated by the software Image-Pro Plus. One can see that when Ce is added, Al_4_Ce particles in bulk shape with a size of below 10 μm show up. Small amounts of Al_4_Ce particles distributed along the ED in streamline can be seen in the alloy containing 0.2 wt % Ce. When the Ce content is 0.8 wt % or higher, the Al_4_Ce particles are well-dispersed in the Mg matrix. Meanwhile, a gradual rise in the area fraction of the Al_4_Ce phases can be observed. During hot extrusion, Al_4_Ce particles can act as nuclei for the DRX, and so it is verified that the increase in the amount of the thermodynamically stable Al_4_Ce particles with an increasing content of Ce addition can provide more nucleation sites for the DRX grains, and thus achieve grain refinement [18]. The excellent grain-refining effect of the Ce addition can also be concluded in the results depicted in Figure 2a–d. 

Engineering tensile stress–strain curves of the as-extruded AZ80-*x*Ce (*x* = 0, 0.2, 0.8 and 1.4 wt %) alloys at room temperature are presented in Figure 5a. The detailed tensile properties are summarized in Table 2. With the addition of Ce, a slight increase for both the yield strength (YS) and ultimate tensile strength (UTS) was obtained, i.e., from ~156.4 MPa and ~305.3 MPa for the AZ80 alloy, to ~180.4 MPa and ~326.2 MPa for the AZ80-0.8Ce alloy, respectively. When the Ce content climbed to 1.4 wt %, both YS and UTS reduced slightly. This phenomenon can be attributed to the reason that when Ce addition rises from 0.8 wt % to 1.4 wt %, the volume fraction of the Al_4_Ce particles increases greatly (Figure 4o–p). These high-melting-point hard phase particles break up during hot extrusion, and their edges and corners become sharp, which can lead to stress concentration and become the source of microcracks, thus reducing strength. However, the grain size of the alloy with the addition of 1.4 wt % Ce was much finer (Figure 2c–d); the strength still remained at a relatively high level, due to the effect of grain refinement strengthening. In contrast, there appeared an obvious and persistent increase for the elongation value from 22.1% for the AZ80 alloy, to 33.9% for the AZ80-1.4Ce alloy. The significant enhancement in the elongation of the Ce-containing extruded alloys is mainly a result of grain refinement, as shown in Figure 2a–d. The reason for such a shift lies in the fact that grain refinement can suppress the nucleation and propagation of cracks by forming more serrated grain boundaries, and reducing the size of possible flaws that may cause stress concentration [19]. Besides, grain refinement can also make the deformation more uniform. Therefore, a AZ80-1.4Ce alloy with the finest grain size can endure the greatest amount of deformation before fracture.

The work-hardening rate (*Θ* = ∂*σ_true_*/∂*ε_true_*) vs. the true strain curves for the uniform plastic deformation stage in tensile tests of the as-extruded AZ80-*x*Ce (*x* = 0, 0.2, 0.8 and 1.4 wt %) alloys are shown in Figure 5b. One can see that the value of the work-hardening rate *Θ* decreases rapidly after yielding. The declining rates obviously slows down obviously at a true strain of ~7%. The *Θ* value of the AZ80 alloy begins to plunge rapidly again, when the true strain comes to ~17%. In comparison, the critical true strain values where the declining rate of *Θ* values speed up again gradually increase with increasing Ce content. When the Ce content is 1.4 wt %, the critical true strain rises to ~27%. Furthermore, AZ80-1.4Ce alloy always shows the highest work-hardening rate at the same true strain after yielding among all of these alloys. These results coincide with the phenomenon observed in Figure 4a, that a higher Ce content contributes to a larger elongation of the as-extruded AZ80-*x*Ce (*x* = 0, 0.2, 0.8 and 1.4 wt %) alloys. The relationship between the true flow stress and the true strain can be described by the Hollomon equation (*σ_true_* = *Kε_true_^n^*), where *n* is the work-hardening index, and *K* is the strength coefficient. The value of the work-hardening index *n* can reflect the uniform deformation ability of metal materials. Generally, higher *n* values always come with a better formability of alloys at room temperature [20]. Based on the Hollomon equation, the average *n* values of the as-extruded AZ80-*x*Ce (*x* = 0, 0.2, 0.8 and 1.4 wt %) alloys are calculated to be 0.34, 0.36, 0.37, and 0.41, respectively. It demonstrates that the addition of Ce makes the AZ80 alloy more capable of deformation. Researchers have reported a dynamic competitive relationship between the dislocation accumulation caused by plastic deformation, and the dislocation annihilation induced by dynamic recovery, which affects the work-hardening capability of as-extruded alloys [21]. As shown in the insets of Figure 4f–h, Al and Zn elements are mostly dissolved into the Mg matrix in the as-extruded alloys. The large amount of solute atoms can remarkably reduce the dynamic recovery rate of the dislocations, and is beneficial for work-hardening. On the other hand, when the Ce content is 0.8 wt % or higher, a large amount of well-dispersed fine Al_4_Ce particles formed in the Ce-containing as-extruded alloys can hinder the recovery of dislocations, which also improves the work-hardening ability. Consequently, as-extruded AZ80-0.8/1.4Ce alloys can achieve an extraordinary combination of high ductility and a relatively high strength.

Figure 6 shows the results of EBSD analyses of the as-extruded AZ80-*x*Ce (*x* = 0, 0.2, 0.8 and 1.4 wt %) alloys at a cross-section paralleled to the ED. From the orientation maps shown in Figure 6a–d, the grain size of the as-extruded AZ80 alloy is greatly refined with the addition of Ce, which is in accordance with the results that the OM images express (Figure 2), as well as the results of the SEM micrographs (Figure 4). A typical sheet texture can be observed in all samples, and the as-extruded AZ80 alloy shows a relatively stronger basal texture. The texture peak intensity at the {0001} pole figure of the as-extruded AZ80 alloy is 18.32 multiples of a random distribution (MRD). It is gradually weakened after adding Ce to the alloy, and it declines to 11.50 MRD when the content of Ce is 1.4 wt %, which corresponds to the results that are indicated in the XRD images (Figure 3). Besides, the texture slightly spreads out from the {0001} plane with the addition of Ce. This phenomenon is well consistent with the extruded RE-containing Mg alloys that are mentioned in [22,23,24], which have weaker textures than those of conventional Mg alloys. Stanford et al. [25] found that a weaker texture could significantly improve the ductility. The responsible reasons are as followed: the main deformation mechanism of Mg alloys at room temperature is basal slip, and the extrusion textures formed in the Mg alloys can make it difficult for basal slips to operate during tensile deformation in the ED, because the basal poles are aligned with the sheets’ normal direction (ND). Consequently, the weakened texture leads to reduced barriers for basal slip, and the deformability of the alloys will also be ameliorated. The distribution histograms of the Schmid factors for basal slips are shown in Figure 6e–h, and the average value of the Schmid factor increases from 0.232 (for AZ80) to 0.293 (for AZ80-1.4Ce), which is consistent with the phenomena that finer grains always come with higher Schmid factors for basal slips [16,26]. Besides, with the addition of Ce, the area fraction of the proportion with a Schmid factor value that is greater than 0.4 significantly surges from 6.6% (for AZ80) to 17.7% (for AZ80-0.2Ce), and then the variation becomes relatively gentle, with area fractions of 20.6% (for AZ80-0.8Ce) and 21.3% (for AZ80-1.4Ce), respectively. So far, many reports have revealed that a higher Schmid factor is usually accompanied with better formability in Mg alloys [26,27,28]. The Schmid factor (*m*) can be calculated by the following equation [29]: *m* = cos *θ* × cos *λ*, where *θ* and *λ* are the angles between the stress direction and the slip direction and the normal direction of the slip plane, respectively. Thus, Schmid factor (*m*) is bounded by 0 and 0.5, and the value of *m* approaches 0.5 when *θ* and *λ* are close to 45°, in which condition the slip system is in the orientation of easy slip, also known as the soft orientation. In this work, the orientations with *m* values ranging between 0.4 and 0.5 are regarded as soft orientations, and the area fractions of the basal slip systems in soft orientations are calculated. As shown in Figure 6e–h, the area fraction rises with increasing Ce content, which is in favor of facilitating dislocation slips and promoting the formability of the as-extruded AZ80 alloy. Figure 6i–l shows the distribution histograms of the misorientation angle. One can find that the fraction of low angle grain boundaries (LAGBs) of the as-extruded AZ80 alloy is 14.4%, while it is 12.3%, 10.0%, and 9.1% in alloys with additions of 0.2 wt.%, 0.8 wt.%, of 1.4 wt.% Ce, respectively. For microcrystalline alloys, plastic deformation is dominated by both intragranular and intergranular dislocation motions [30]. High-angle grain boundaries (HAGBs) with misorientation angles larger than 15° exhibiting relatively lower thermal and mechanical stabilities, and providing barriers for the transmission of dislocations from one grain to another, then creating dislocation pile-ups at the boundaries and enhancing the macroscopic strength [31]. In addition, when cracks occur in grains, a larger number of HAGBs are responsible for better crack propagation resistance [32]. On the contrary, LAGBs can even be described as an array of dislocations; thus, cracks can be transferred directly through the LAGBs into other grains. In conclusion, the reduction of the fraction of LAGBs can suppress the propagation of cracks, and it is beneficial for mechanical properties.

In order to further identify the phases in the as-extruded alloys, TEM analyses were performed with an as-extruded AZ80-0.8Ce alloy. Bright-field TEM micrographs are presented in Figure 7a,c,e,g,i. Corresponding diffraction patterns acquired from selected regions marked by the arrows A, B, C, and D in Figure 7a,c,e,g are shown in Figure 7b,d,f,h, respectively. One can see that plate-like and block-shaped Mg_17_Al_12_ precipitates (region A and B) with sizes of 300–600 nm, disperse in the AZ80-0.8Ce alloy. Coarse tabular-blocky Al_4_Ce phases (region C) with sizes of 500–700 nm, as well as fine rod-like and spherical Al_4_Ce phases (region D) with sizes of 50–200 nm can also be observed. These results are in good agreement with the XRD results shown in Figure 3c. It is well known that Mg_17_Al_12_ precipitates are strengthening phases in the extruded AZ80 alloys, which can be formed by dynamic precipitation from a supersaturated solid solution during hot extrusion [33]. However, adding Ce can restrain the precipitation of Mg_17_Al_12_ particles, since the formation of a thermodynamically stable Al_4_Ce phase would consume Al. Fine nano-sized Al_4_Ce particles can precipitate during the hot extrusion process, while irregular bulk-like submicron-/micron-sized Al_4_Ce particles may be formed from the primary Al_4_Ce, emerging in a smelting process and breaking in the extrusion process. Figure 7i shows a typical fine rod-like Al_4_Ce particle in the *α*-Mg matrix. A high-resolution TEM (HRTEM) image of zone F in Figure 7i is shown in Figure 7j. A clear interface between the nano-Al_4_Ce particle and the *α*-Mg matrix can be observed, illustrating good interfacial bonding, which is beneficial for transferring the load from the Mg matrix to the particles, and thus improving the strength of the alloys. Figure 7k presents the inverse fast-Fourier-transform (IFFT) image of zone E in Figure 7j. Apparently, a large number of dislocations are distributed in the *α*-Mg matrix adjacent to the nano-Al_4_Ce particle. A high dislocation density is reported to be beneficial to the precipitation of nano-Al_4_Ce particles as heterogeneous nucleation sites [34]. Besides, the nano-Al_4_Ce particles could also act as strong obstacles to the dislocation motion and result in a strengthening effect. However, on account of the incomplete solid solution of Al_4_Ce, the number of nano-Al_4_Ce particles is quite limited, and the effect of precipitation reinforcement is insignificant.

## 4. Conclusions

In the present study, the effect of Ce content on the microstructure and tensile properties of as-extruded AZ80-xCe alloys (*x* = 0, 0.2, 0.8 and 1.4 wt %) were investigated. In the AZ80 alloy, two phases could be observed, including the primary *α*-Mg phase and the Mg_17_Al_12_ phase. With the addition of Ce, a bulk-shaped Al_4_Ce phase below 10 μm appeared, and a small number of Al_4_Ce particles were distributed along the ED for the AZ80-0.2Ce alloy. When the Ce content was 0.8 wt % or higher, Al_4_Ce particles in both nano- and micron sizes were well dispersed in the *α*-Mg matrix. The area fraction of Al_4_Ce particles rose with increasing Ce content, indicating a larger number of thermodynamically stable Al_4_Ce particles, which could provide more nuclei for DRX during hot extrusion, and promote grain refinement. The grain size of the as-extruded AZ80-1.4Ce alloy was significantly reduced by ~56.7%, compared to that of the AZ80 alloy. Adding Ce could lead to significant grain refinement, a weakened texture strength, an increased Schmid factor, and a reduced area-fraction of LAGBs. All of these were beneficial to improving the deformability of the extruded AZ80 alloy. With Ce content increasing to 0.8 wt % or higher, a superior combination of tensile properties could be achieved, with a high elongation of ~29–34%, and a relatively high ultimate tensile strength of ~320–327 MPa, respectively.

## Figures and Tables

**Figure 1 materials-12-00076-f001:**
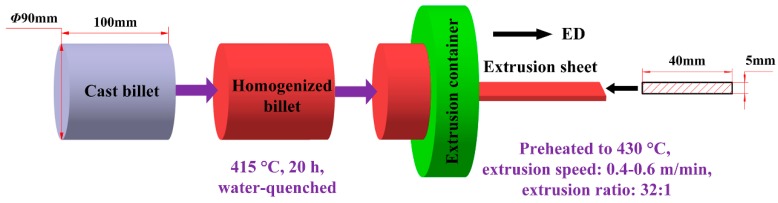
The schematic diagram of the extrusion process in this work.

**Figure 2 materials-12-00076-f002:**
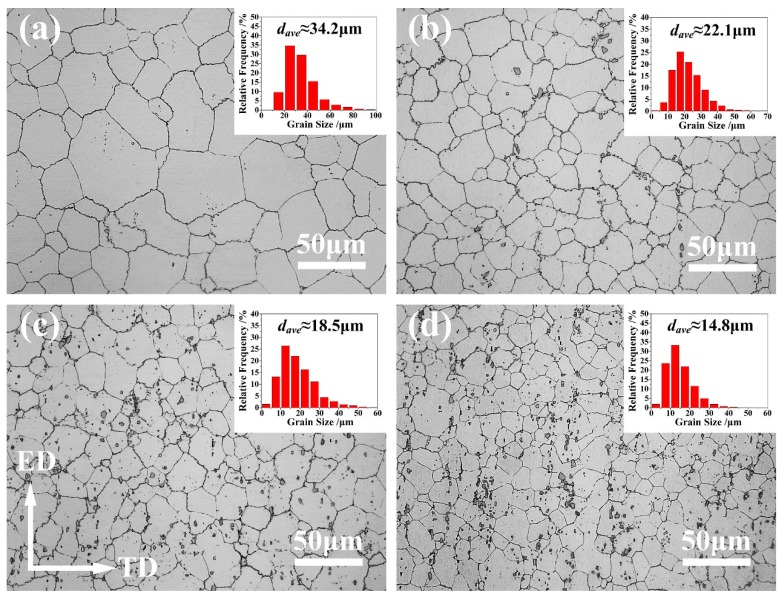
The optical microscope (OM) images and the grain size distribution histograms (inset: *d_ave_* stands for the average grain size) of the as-extruded AZ80-*x*Ce alloys: (**a**) *x* = 0; (**b**) *x* = 0.2; (**c**) *x* = 0.8; (**d**) *x* = 1.4. ED and TD stand for extrusion direction and transverse direction, respectively.

**Figure 3 materials-12-00076-f003:**
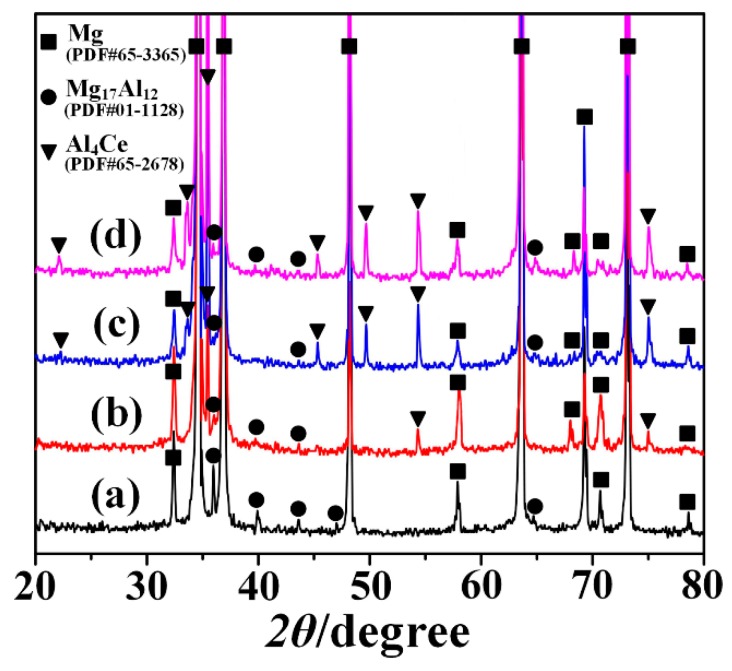
The X-ray diffraction spectra of the as-extruded AZ80-*x*Ce alloys: (**a**) *x* = 0; (**b**) *x* = 0.2; (**c**) *x* = 0.8; (**d**) *x* = 1.4.

**Figure 4 materials-12-00076-f004:**
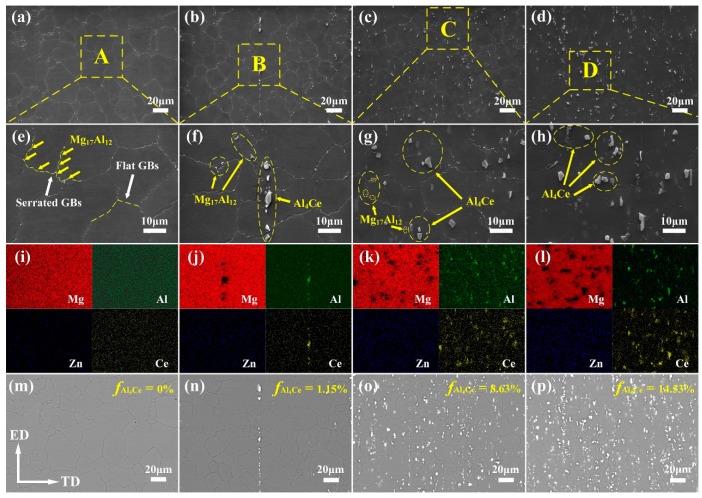
(**a**–**d**) Scanning electron microscope (SEM) micrographs of the as-extruded AZ80-*x*Ce alloys: (**a**) *x* = 0; (**b**) *x* = 0.2; (**c**) *x* = 0.8; (**d**) *x* = 1.4; (**e**–**h**) corresponding magnified SEM diagrams of the selected zone A, B, C, and D in (**a**–**d**), respectively; (**i**–**l**) corresponding energy dispersive spectrometer (EDS) mapping images of (**e**–**h**) (remark: each image contains four parts to show the different elements, Mg, Al, Zn, and Ce are marked in red, green, blue and yellow, respectively); (**m**–**p**) corresponding backscattered electron (BSE) micrographs of (**a**–**d**) (remark: *f*Al_4_Ce indicates the area fraction of Al_4_Ce phases).

**Figure 5 materials-12-00076-f005:**
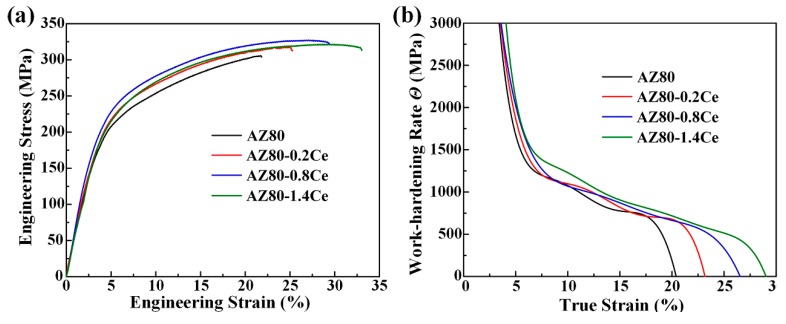
(**a**) Tensile engineering stress–strain curves of as-extruded alloys; (**b**) corresponding work-hardening rate *Θ* vs. the true strain of (**a**).

**Figure 6 materials-12-00076-f006:**
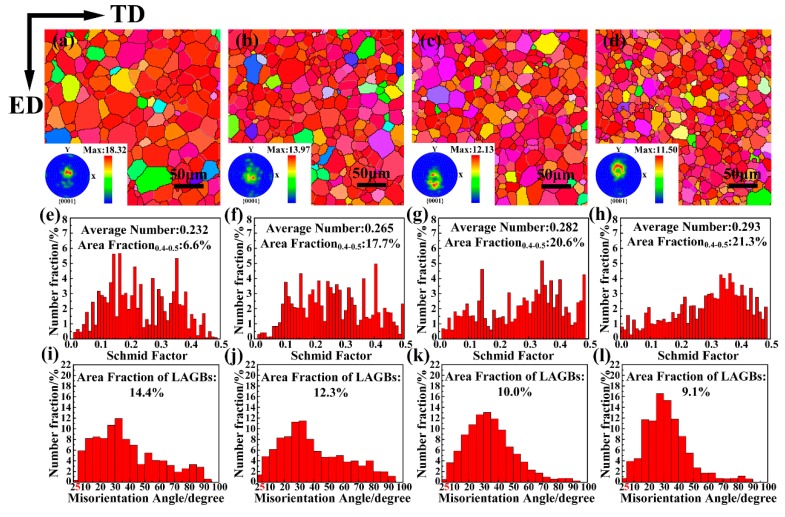
(**a**–**d**) The electron backscattered diffraction (EBSD) orientation maps and the corresponding {0001} pole figures; (**e**–**h**) the basal slip Schmid factor distribution histograms; (**i**–**l**) the misorientation angle distribution histograms (remark: LAGBs stands for low angle grain boundaries with misorientation angles that are larger than 2° but smaller than 15°) of the as-extruded AZ80-*x*Ce alloys: (**a,e,i**) *x* = 0; (**b,f,j**) *x* = 0.2; (**c,g,k**) *x* = 0.8; (**d,h,l**) *x* = 1.4.

**Figure 7 materials-12-00076-f007:**
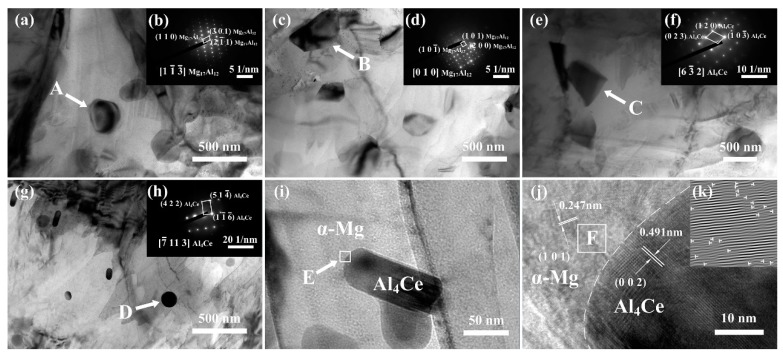
(**a,c,e,g,i**) Bright-field transmission electron microscope (TEM) micrographs of the as-extruded AZ80-0.8Ce alloy; (**b,d,f,h**) corresponding diffraction patterns of selected particles taken from the (**a,c,e,g**), indicated by arrows of A, B, C and D, respectively; (**j**) high-resolution TEM (HRTEM) image of the zone E in (**i**); (**k**) inverse fast-Fourier-transform (IFFT) image of the zone F in (**j**) (remark: the “T” shaped symbols represent for the dislocations).

**Table 1 materials-12-00076-t001:** The chemical compositions of the measured alloys.

Nominal Composition	Measured Composition (wt %)			
	Al	Zn	Ce	Mg
**AZ80**	$7.90_{-0.06}^{+0.05}$	$0.55_{-0.04}^{+0.01}$	-	Bal.
**AZ80-0.2Ce**	$7.92_{-0.07}^{+0.05}$	$0.53_{-0.02}^{+0.01}$	$0.20_{-0.01}^{+0.01}$	Bal.
**AZ80-0.8Ce**	$7.81_{-0.03}^{+0.01}$	$0.52_{-0.01}^{+0.01}$	$0.79_{-0.02}^{+0.01}$	Bal.
**AZ80-1.4Ce**	$7.73_{-0.16}^{+0.10}$	$0.51_{-0.01}^{+0.01}$	$1.34_{-0.03}^{+0.02}$	Bal.

**Table 2 materials-12-00076-t002:** Tensile properties of the as-extruded AZ80 alloys with different Ce.

Composition	Yield Strength σ_0.2_ /MPa	Tensile Strength σb /MPa	Elongation εb /%
**AZ80**	$156.4_{-2.7}^{+3.6}$	$305.3_{-5.8}^{+4.3}$	$22.1_{-2.1}^{+1.3}$
**AZ80-0.2Ce**	$162.1_{-0.5}^{+1.6}$	$314.9_{-4.0}^{+2.5}$	$24.3_{-1.3}^{+1.2}$
**AZ80-0.8Ce**	$180.4_{-4.2}^{+3.1}$	$326.2_{-1.4}^{+1.0}$	$29.1_{-1.4}^{+0.8}$
**AZ80-1.4Ce**	$161.1_{-2.7}^{+4.7}$	$320.7_{-0.4}^{+0.8}$	$33.9_{-0.5}^{+1.1}$

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
