# Peer review of "Effect of Ce Addition on Modifying the Microstructure and Achieving a High Elongation with a Relatively High Strength of As-Extruded AZ80 Magnesium Alloy"

_materials, 2018, doi:10.3390/ma12010076_

Round 1

Reviewer 1 Report

Dear Authors,

Following are some remarks to the work:

1.      A lot of works have been made on investigation of Mg alloys modification, moreover, AZ80 alloy modification by Sr and Sn was well described. Can you clarify the novelty of the work and why it is essential?

2.      Abstract is too long and contains a lot of info, please take a look again and delete not useful info.

3.      Table 1 presents some values, have you tested the alloy once? If no, explain how many measurements have you made and add standard deviation to the values.

4.      Please explain what the reason to perform XRD investigation was.

5.      Figure 4: mapping images are too small, nothing can be recognized for the reader please enlarge them.

Author Response

Thank you very much for your comments and suggestions. They are very valuable in improving the quality of our manuscript. We have carefully revised our original manuscript according to your comments and advices. The attached file is a point-to-point response to your comments.

Reviewer 2 Report

For the Authors

The studies of the Ce addition effect on the microstructure evolution and tensile properties of the as-extruded AZ80-xCe alloys are presented in the article. The authors obtained interesting research results. However, there are some comments and recommendations.

1.      As a result of the analysis of XRD-patterns, it can be concluded that with increasing of Ce content, the intensity of the reflexes corresponding to the phase Al4Ce increases. In this case, the intensity of the Mg17Al12 phase reflexes decreases. However, at the same time there is no comments and explanations for changes occurring in the structure of magnesium. Some of magnesium reflexes increase with the Ce concentration increasing for example for the angles of 33 and 58 degrees. Some new reflexes of magnesium appear for example for the angle of 68 degrees. How the authors explain this phenomenon?

2.      It should be noted, that an unidentified reflex appears in the area of the angle of 34 degrees (Fig. 3c,d).

3.      Element distribution maps are located unsuccessfully on SEM images and are not clear enough. The elements distribution map for area A is not shown. Although it should be given for comparative analysis. There is no the quantitative element analysis.

4.      The authors did not explain why when the Ce concentration increases from 0.8 to 1.4, the Yield strength and Tensile strength decrease. Probably this is due to the phase composition changing. Detailed comprehensive analysis of the results is necessary in this case.

5.      The submitted manuscript can be accepted for the publication in “Materials” after the minor revisions.

Author Response

(The authors gave the same response as above.)

Reviewer 3 Report

In my opinion, the manuscript is well written and contains interesting research, I recommend it for publication in Materials after minor revision.

I found a few minor mistakes in the text:

-          The authors did not specify the manufacturer of the use of chemical reagents.

-          Line 103 – “20h” no space.

-          Fig.2 (inside) – the descriptions overlap with bar charts.

-          Fig 4. EDS mapping images are they too small and illegible.

-          In the work there is a double sign with the symbol θ (first in fig.1 and second line 261, 263), this should be improved.

-          Fig.6e, g, l – the descriptions overlap with bar charts.

-          The all symbols and constants should be written in italics (in text and in figures) for examples: lines 53, 84, 92, 144, 146, (fig.2), 155, 156, (fig.3), 158, 159, 161, 162, 164, 165, 187, 208, 217, 218, 219, 222, 235, 260, 261, 263, 264, 287, 321.

Author Response

(The authors gave the same response as above.)

Reviewer 4 Report

Dear Authors,

I will recommend the article for publishing after you address my minor remarks:

1.       Keywords – I do not like keyword „Ce” because it is too confusing. I suggest “Ce modification” or something like that.

2.       Line 40, 41 and similar cases: “Magnesium alloys”- I do not feel the capital letter is necessary.

3.       Figure 6: “Misorietation” – should be “Misorientation”

Sincerely,

Reviewer

Author Response

(The authors gave the same response as above.)

Reviewer 5 Report

It is an interesting study that adds new data to forming magnesium alloys with rare earth elements. The topic and scope of the manuscript is highly suitable for the readership of the Materials. I recommend a minor revision of the paper only.

Some detailed comments are as following:

1). In Materials and Experimental Procedures, it is necessary to cite earlier works by the authors or the work of other authors from the literature on the applied temperature conditions of thermal treatment of the alloys under test. Please, correct.

2). What method was used to determine the chemical composition of the tested alloys? Table 1 shows the lack of standard deviation values for the measured composition. Please, correct.

3). In Materials and Experimental Procedures, line 135, the sentence: „Tensile specimens with a gauge 134 size of 10 mm × 3.5 mm × 1.8 mm…” should be changed into Tensile specimens with a gauge 134 size of 10 × 3.5 × 1.8 mm…”. All values should be given in mm and the length unit should be given only once. Please, correct.

4). The determined grain size was given in the text without any standard deviation value. Please, correct.

5). In Results and Discussion, the Authors have to add the numbers of ICDD PDF card used for the phase identification of the tested materials.

6). In the caption to Figure 3, the XRD abbreviation was unnecessarily re-entered. Please delete the XRD abbreviation because it has already been entered in the text.

7). Individual sub-points should not end with drawings and tables, but with a text describing the results previously presented (see Figure 7). Please, correct the order.

Author Response

(The authors gave the same response as above.)

Round 2

Reviewer 1 Report

Dear Authors,

Thank you for revising the manuscript according to my comments.